# Microfluidic Systems for Neural Cell Studies

**DOI:** 10.3390/bioengineering10080902

**Published:** 2023-07-30

**Authors:** Eleftheria Babaliari, Anthi Ranella, Emmanuel Stratakis

**Affiliations:** 1Foundation for Research and Technology—Hellas (F.O.R.T.H.), Institute of Electronic Structure and Laser (I.E.S.L.), Vasilika Vouton, 70013 Heraklion, Greece; ebabaliari@iesl.forth.gr; 2Department of Physics, University of Crete, 70013 Heraklion, Greece

**Keywords:** neural cells, neural tissue engineering, microfluidics, microfluidic flow, shear stress, topography

## Abstract

Whereas the axons of the peripheral nervous system (PNS) spontaneously regenerate after an injury, the occurring regeneration is rarely successful because axons are usually directed by inappropriate cues. Therefore, finding successful ways to guide neurite outgrowth, *in vitro*, is essential for neurogenesis. Microfluidic systems reflect more appropriately the *in vivo* environment of cells in tissues such as the normal fluid flow within the body, consistent nutrient delivery, effective waste removal, and mechanical stimulation due to fluid shear forces. At the same time, it has been well reported that topography affects neuronal outgrowth, orientation, and differentiation. In this review, we demonstrate how topography and microfluidic flow affect neuronal behavior, either separately or in synergy, and highlight the efficacy of microfluidic systems in promoting neuronal outgrowth.

## 1. Introduction

Taking into consideration the fact that neurological injuries are hardly self-recoverable, the development of successful methods to guide neurite outgrowth, *in vitro*, is of high significance. Indeed, although the peripheral nervous system (PNS) exhibits a higher rate of regeneration than that of the central nervous system (CNS) through spontaneous regeneration after injury, functional recovery is fairly infrequent and misdirected [1].

Neural tissue engineering has emerged as a promising alternative field for the development of new nerve graft substitutes to overcome the limitations of the current grafts [2]. Indeed, the ultimate goal of a tissue-engineered construct is to sufficiently mimic the topographic features of the extracellular matrix and the surrounding environment of cells (e.g., mechanical properties, soluble factors, shear stress) so that cells will respond in their artificial environment in the same way they would *in vivo*.

Several approaches have been developed to create cellular substrates at the micro- and/or nano-scales that aim to reconstruct the architecture of the extracellular matrix *in vitro* [3,4,5,6]. However, apart from topography, it has recently become increasingly evident that neurogenesis may be also driven by mechanical factors [7,8]. Indeed, shear stress is a significant component of the host environment of regenerating axons [9]. Flow-induced shear stress can be applied to cells, *in vitro*, using specially designed microfluidic systems. In such systems, microfluidic flow replicates the physiological fluid flow inside the body, facilitates mass transport of solutes, and supplies consistent nutrient delivery and effective waste removal resulting in a more *in vivo*-like environment [10]. Previous research and review studies have described the main fabrication techniques, the produced types of microfluidic systems, as well as their respective advantages and disadvantages [10,11,12,13,14,15,16].

In this review, we aim to demonstrate how topography and microfluidic flow affect neuronal behavior, either separately or in combination, with the aid of microfluidic systems, and highlight the importance of taking into consideration many different stimuli (e.g., topography, microfluidic flow, shear stress) to promote neuronal outgrowth. Specifically, in the first part of this review, we show the importance of topography, shear stress, and microfluidic flow on cells’ function, while we also present some limitations of cell culture under microfluidic flow. Finally, in the second part, we review the existing body of literature on the application of microfluidic systems for neural studies focused on the gradient—chemical stimulation, perfusion—shear stress, and co-culture, as well as the application of microfluidic systems in combination with topography for neural studies.

## 2. Neuroregeneration—Tissue Engineering

Human adult nerve injuries are a major clinical issue that significantly affects the quality of patients’ lives [17], causing 6.8 million deaths annually and affecting the lives of over 1 billion people worldwide [18]. These injuries have been correlated with a wide range of disorders such as brain and spinal cord traumatic injuries, neurodegenerative diseases, and stroke [19].

The nervous system is subdivided into the central nervous system and the peripheral nervous system. Although the central nervous system rarely regenerates after an injury, the peripheral nervous system has an intrinsic regenerative capacity [20]. If a peripheral nerve is injured and its axons are severed, the parts of the axons distal to the damage site are cut off from the central cell body. Afterward, the debris generated by the degradation of these distal axon segments is removed by Schwann cells and macrophages in a process called “Wallerian degeneration”. Proliferating and adopting a pro-regenerative phenotype, Schwann cells reorganize into tracts in alignment known as “bands of Büngner” and secrete factors that promote the development of regenerated axons. Meanwhile, the damaged neurons’ parts close to the injury site prepare for regeneration. Each damaged axon’s cell body experiences significant metabolic changes and a protein production program is started to aid in the regeneration of axons. Every damaged axon generates a growth cone at the tip that guides the regenerating axon through the distant nerve segment’s undamaged structure and back toward its original target at a rate of roughly 1–3 mm each day [20,21]. However, the regeneration of peripheral nerves occurs spontaneously only if the injury does not cause a significant gap in the nerve. Most of the time, such regeneration is rarely functional because when a nerve fiber loses its continuity with consequent damage of the basal lamina tubes, axon spontaneous regeneration is disorganized and mismatched, resulting in an inadequate nerve functional recovery and musculoskeletal incapacity [17].

Nerve autografts remain the gold standard in the treatment of peripheral injuries. From the patient’s body, healthy nerve segments are removed and used to close the gap across the damaged site. However, this approach has some disadvantages including the necessity for subsequent surgery, the loss of a functional nerve, donor site morbidity, and longer surgical procedures [17,20]. Therefore, tissue engineering has emerged as a promising alternative field for the development of new graft substitutes [2]. Indeed, the ultimate goal of using a tissue-engineered construct is to adequately mimic both the topographic features of the extracellular matrix (ECM) as well as its surrounding stimuli, including for example mechanical stresses and soluble factors, so that cells will respond within the artificial environment as they would *in vivo*.

In tissue-engineered constructs, the effective utilization of biomaterials permits spatiotemporal control of environmental cues enabling successful regeneration and repair of neuronal deficits. Moreover, biomaterials are used in tissue engineering techniques to transplant cells that replace or augment the functionality of dead or diseased neural cells. Indeed, cell transplantation scaffolds may contain cues that could affect both the transplanted cells, as well as the host neural tissue of the implantation site [20].

For the aforementioned reasons, the development of a scaffold as an alternative to an autograft has drawn considerable scientific interest. The ultimate goal is to develop scaffolds or cell substrates that could be implanted into the injury gap in lieu of a nerve graft to guide and support regenerated cells and axons. A construct like this would employ biomaterials to mimic the pro-regenerative environment found in the distal part of the damaged nerve. For instance, a physically aligned surface topography with oriented tracks of adhesive molecules is provided by the Büngner bands that are present in the distal nerve segment. Moreover, cells in the distal nerve stump release soluble biochemical growth and signaling factors. Thus, the development of scaffolds mimicking these kinds of chemical and physical cues in order to direct regenerating axons and migrating cells through the nerve gap and into the distal nerve segment is of great importance [20].

## 3. Importance of Size Scale—Topography

The micro/nanoscale of the cell environment has a significant impact on how the cells respond to their surrounding topography. Microscale features perform at the cellular level, whereas submicron and nanoscale features perform at subcellular levels [22]. Growth cone filopodia, for example, are a type of cell structure that is in the nanometer range and is extremely responsive to topographic cues at the nanoscale [23]. In addition, a dense lattice of the three-dimensional topography, including fibers and pores with dimensions ranging from tens to hundreds of nanometers, makes up basement membranes, across which axons normally develop [24,25]. Consequently, control over micro/nanotopography is considered essential.

At the interface between cells and materials, all the cellular processes are influenced by physical and chemical stimuli originating from the substrate topography, stiffness, and chemistry. Simultaneously, at the intracellular level, focal adhesions play a crucial role as molecular complexes responsible for sensing the surrounding environmental conditions, serving as important mechanosensitive players [26,27,28,29]. In fact, many studies showed that surface topography influences the adhesion, polarization, migration, proliferation, and differentiation of cells [30,31,32,33,34,35,36,37].

In particular, during cell development and growth, neural cells react strongly to naturally occurring stimuli that are present in their surroundings. For instance, a regenerated axon’s path is directed both by the chemistry and physical topography of the surface along which it develops and by signaling molecules in their micro/nano environment. Such physicochemical cues have a significant impact on cellular viability, activity, and growth [20].

Along with structural support at the tissue level, scaffolds with patterned surfaces also offer subtle topographic cues. Controlling the distribution and nature of these cues offered by the scaffolds could influence the attachment, growth, migration, alignment, and gene expression of cells [20].

Due to their sensitivity to their naturally occurring topographic cues, cells have the capacity to react to artificial topographic cues. For instance, aligned ECM is crucial in directing neural cell migration and differentiation during the development of neural tissue [38]. Engineered substrates with specific topographies may have a significant impact on cellular behavior by imitating their natural environment along which neural cells grow.

The interactions between the cellular structures and their substrate topography determine cell behavior through complicated mechanisms. Specifically, neural growth cones found at the expanding axons/neurites are highly sensitive to topographical cues since they guide neuronal extension during cell development and growth. Moreover, filopodia protruding from the ends of growth cones continuously extend or retract in response to chemical and physical cues as they “explore” their surroundings. Growth cones also mediate neuronal cytoskeletal reorganization which in turn affects the direct axon/neurite extension [20].

Changes in the orientation of cells are governed by cytoskeletal reorganization, which in turn controls cell migration and growth. In particular, by depolymerizing and polymerizing the cytoskeleton’s microtubules, intermediate filaments, and microfilaments, the cytoskeleton is reorganized [39]. It is assumed that these rod-like structures have a limited capacity to bend, like a railroad track, and as a result, the surrounding topography affects their polymerization. This perhaps explains why cell growth and alignment can be directionally constrained by specific types of topographic cues like aligned grooves.

Consequently, size scale and topography are of high importance for the design and fabrication of advanced biomaterials in neural tissue engineering. This task is frequently difficult, though, since natural cues are provided in complicated combinations within the intricate terrain of neural tissue. Thus, the surface modification of materials as platforms for studying cell viability, motility, differentiation, and apoptosis is the subject of extensive research [40,41,42].

Several approaches, including femtosecond laser structuring, photolithography, soft lithography, and electrospinning, have been demonstrated to create structures on surfaces that aim to mimic the extracellular matrix *in vitro.* Previous reports and review studies have described the main fabrication techniques, the manufactured micro/nanostructured substrates, as well as the respective advantages and disadvantages [18,40,41,43].

## 4. Role of Shear Stress on Cells’ Functions

Shear stress is termed as “the force per unit area that is created when a tangential force acts on a surface” [44] (Figure 1).

Apart from topography, mechanical stress is also a significant component of the host environment, as it influences the cellular signal transduction and the behavior of various cells. Flow-induced shear stress, in particular, influences mechanoreceptors, like ion channels and integrin/focal adhesions, as well as responses, such as nitric oxide and intracellular calcium production, and cytoskeletal remodeling [46]. Shear stress is applied at discrete local points and is transmitted through the cell body along cytoskeletal microstructures, which in turn trigger intracellular mechanical signaling. Thus, shear stress alters not only the cell’s shape but also the intracellular signaling pathways [1,47] (Figure 2).

Flow-induced shear stress can be applied to cells, *in vitro*, using specially designed microfluidic systems. In particular, fluid flow applies shear stress to a monolayer of cultured cells (Figure 3). Assuming parallel-plate geometry, the applied shear stress is often approximated by the equation of wall shear stress [48,49]:$$\tau_{w}=\left(6\mu Q\right)/(bh^{2})$$
where μ is the fluid viscosity, Q is the flow rate, b is the width of the parallel plates, and h is the separation (height) of the parallel plates.

Previous studies have shown that fluid-induced shear stress enhances the cells’ alignment, through the reorganization of the cytoskeleton, in a variety of cell types [50,51,52,53,54,55,56]. An association between extracellular matrix alignment and cell shear stress was also reported in [57]. In addition to this, it has been shown that such stimulation affects cellular migration [52].

Chafik et al. [9] reported that shear stress is a critical component of the natural environment for the regeneration of axons. Indeed, it is known that the cell soma and the neurites of neurons are correlated to the cytoskeleton, which senses the mechanical stimuli producing different cellular responses [1,9,58]. Moreover, it has been reported that cell proliferation, migration, and differentiation are controlled via numerous unknown signaling pathways connected to the cytoskeleton [58,59]. Thus, fluid-induced shear stress may be also crucial for guiding neurite outgrowth.

## 5. Cell Response under Microfluidic Flow

Over the last decades, conventional cell culture techniques have been well established and essentially consist of culturing cells in flasks, Petri dishes, and microtiter plates [15]. The cells are usually maintained under static conditions and have limited cell-cell interactions. However, compared to traditional static cultures, the environment in which cells in a multicellular organism live is significantly different. They are surrounded by nutrients and fluid and attached to softer substrates than the plastic and glass substrates used in the majority of *in vitro* investigations [13,50] (Figure 4). To gain more insight into different biological problems, it is important to perform cellular experiments that reflect more appropriately the *in vivo* conditions with cell–cell, cell–soluble factor, and cell–matrix interactions [60].

Microfluidic devices provide a more realistic environment for biological research, as they are related to scales found in biological systems (micro- and nano-). Cells are in the micrometer range, typically around 10–15 μm. Unlike traditional static cultures, microfluidic cell cultures permit precise control of the microenvironment (e.g., changes in the flow rate, oxygen (O_2_) levels, pH) that influence biochemical and mechanical factors in a cell and, thus, cell functionality [13]. Figure 5 illustrates the possible physicochemical and biomolecular stimuli provided by the microfluidic flow.

In particular, microfluidic flow cell cultures represent more closely the *in vivo* cell environment such as the physiological fluid flow in the body, consistent nutrient supply, effective waste removal, and mechanical stimulation due to fluid shear forces [14]. Small-length scales, laminar flow regimes, and diffusion-dominated mass transport characterize the microfluidic devices resulting in a more *in vivo*-like environment [10]. In fact, the continuous flow of nutrients achieved by microfluidic flow in dynamic cell cultures offers a distinct advantage over conventional static ones. Furthermore, cells cultured in smaller volumes of media in microfluidic devices reflect the physiological state of tissues more accurately than cells cultured in larger volumes do due to faster nutrient consumption and higher concentrations of secreted products and metabolites, similar to densely packed tissues [14,61].

Table 1 summarizes some important differences between conventional static cultures and cell cultures under microfluidic flow.

Utilizing the microfluidic technology discussed above, numerous microfluidic platforms applied to stem cells [62,63,64,65,66], tumor cells [67,68,69], and other types of cells [52,53,54,55,56,70,71,72,73]. Microfluidic platforms are designed for several specific applications including single-cell studies [74,75,76], biomarkers’ detection [77,78,79], drug screening and discovery [80,81,82], organs-on-chip [83,84,85,86,87,88,89,90,91,92], and tissue engineering [10,93,94].

A summary of applications for individual cell types is presented in Figure 6.

## 6. Application of Microfluidic Systems for Neural Studies

The positive effect of dynamic cultures on the response of neuronal and/or glial cells (such as adhesion, proliferation, directionality, and differentiation) has been demonstrated by many studies. In this section, we report studies on the (i) gradient and chemical stimulation of neural cells, (ii) perfusion and shear stress on neural cells, and (iii) co-culture of neural cells via microfluidic systems.

### 6.1. Gradient Generation—Chemical Stimulation

Chung et al. [96] developed a polydimethylsiloxane (PDMS) gradient-generating microfluidic platform that exposed human neural stem cells (hNSCs) to a concentration gradient of known growth factors (GFs), under continuous flow (5 × 10^−5^ Pa). As a result, a minimization of autocrine and paracrine signaling was observed. Additionally, they found that the differentiation of hNSCs into astrocytes was inversely proportional to GF concentration while proliferation was directly proportional (Figure 7A–C). Kim et al. [97] performed a similar study by using a microfluidic chip-generated growth factor gradient system and neural stem cells (NSCs). Results showed that NSCs proliferation and differentiation were directly dependent on the concentration gradient of GF (Figure 7D,E). Additionally, Nakashima and Yasuda [98] fabricated a microfluidic device to investigate the effect of GF on the differentiation and axon elongation guidance of adrenal pheochromocytoma (PC12) cells. The microfluidic device was composed of a cell culture chamber, a micro-channel, a nano-hole array (containing GF), and a micro-valve allowing the precise release of chemicals from the nano-hole. Nerve growth factor (NGF) was used to stimulate the differentiation of PC12 cells. They showed that the cell growth, differentiation, and axon elongation were dependent on micro-valve switching and release gradient of NGF (Figure 7F,G). Futai et al. [99] developed a one-layer H-shaped microfluidic channel, for concentration gradient generation, consisting of a thin (~2 μm) but high-aspect-ratio (0.5–1) microchannel. Using a long-lasting concentration gradient of NGF, they examined the axon elongation of primary dorsal root ganglion (DRG) neuronal cells. The results revealed a directional elongation of axons following the NGF concentration gradient during cultivation for 96 h (Figure 7H,I). Furthermore, Park et al. [100] designed a microfluidic platform to expose human embryonic stem cells (ESCs)-derived neural progenitor cells to stable concentration gradients of extracellular signaling molecules. Human ESC-derived neural progenitor cells were cultured in the microfluidic system under continuous cytokine gradients (0.15 μL/h) for 8 days. Neural progenitor cells proliferated and differentiated, into neurons, in a controlled manner, and the cell properties reflected the different concentrations of extracellular signaling molecules (Figure 7J,K). Bhattacharjee and Folch [101] fabricated a microfluidic chip containing 1024 biochemical gradient generators, with each generator entrapping a single neuron, to investigate the axon guidance and growth dynamics of primary hippocampal neurons in response to biochemical cues. The microfluidic chip produced a reproducible, stable gradient with negligible shear stress on the culture surface (100 μL/h). Using this platform, it was demonstrated that the hippocampal axon guidance was dependent on the concentration and incidence angle of the netrin-1 gradient. In particular, regarding the concentration, it was found that hippocampal growth cones close to the source of netrin-1 (high concentration) were strongly attracted, while those far from the source of netrin-1 (low concentration) were repelled. With regards to the angle of incidence, it was shown that hippocampal growth cones oriented away from the gradient axis (90–135°) turned toward the netrin-1 source, while those oriented toward the gradient (less than 45°) were strongly repelled (Figure 7L,M). Finally, Cheng et al. [102] developed a microfluidic system that could analyze the effect of chemical and mechanical stimulation on the neuronal differentiation of placenta-derived multipotent stem cells (PDMCs). They investigated the effect of shear stress using different flow rates (1.4 × 10^−4^, 3.31 × 10^−3^, 4.97 × 10^−3^ Pa) on PDMCs beside 3-isobutyl-1-methylxanthine (IBMX), as the chemical stimulant, for 3 days. They showed that shear stress could not differentiate PDMCs into other cell types. Although chemical stimulation played a crucial role in the differentiation of PDMCs, shear stress enhanced PDMCs for earlier neuronal differentiation. In the 48-h condition, the maximum flow rate and IBMX showed the highest cell differentiation ratio of 42.4% (Figure 7N–P).

The neuronal response to gradient and chemical stimulation through microfluidic systems is summarized in Table 2. The different approaches and the respective nerve responses are presented.

### 6.2. Perfusion—Shear Stress

Chafik et al. [9] developed a custom-designed flow chamber that applied shear stress (1.33 Pa for 2 h), through laminar fluid flow, to Schwann cells. They showed that mechanical stimuli enhanced the proliferation of Schwann cells and caused a slight movement from their original positions (Figure 8A,B). Gupta et al. [103,104] used an *in vitro* model to apply shear stress on primary Schwann cells in the form of laminar fluid flow (3.1 Pa for 2 h). They observed increased proliferation and downregulation of two pro-myelinating proteins, myelin-associated glycoprotein (MAG) and myelin basic protein (MBP). These results implied that a low level of mechanical stimulus may directly trigger Schwann cell proliferation (Figure 8C–E). Moreover, Millet et al. [105], considering that cell-to-cell signaling is local, developed a specific culture system that sustained small numbers of primary hippocampal neurons and enabled analysis of the microenvironment. They observed that cultured neurons inside perfused channels (by gravity flow) composed of native, autoclave, or extracted PDMS showed increased viability and channel-length capacity (increasing 2-fold for all but native PDMS) (Figure 8F,G). Park et al. [106] designed a two-dimensional microfluidic system to study the effect of continuous flow shear stress (10^−4^ and 10^−3^ Pa) on radial glial cells (RGCs). They found that flow shear stress possibly activated mechanosensitive Ca^+2^ channels that significantly enhanced the proliferative capacity of RGCs in response to increased shear stress (Figure 8H,I). Furthermore, Park et al. [107] developed a microfluidic chip to apply a continuous flow of fluid that is readily observed in the brain’s interstitial space, on three-dimensional (3D) micro-spheroidal neural tissue (neurospheroids). The slow interstitial level of flow (0.15 μL/min) was maintained using an osmotic micropump system and the uniform neurospheroids were formed in concave microwell arrays. Using this system, the effect of flow on the size of neurospheroids, neural networks, and neural differentiation was investigated. Under flow conditions, the results indicated that the size of neurospheroids was larger and formed more complex and robust neural networks (increased levels of synapsin IIa and β-ΙΙΙ tubulin, decreased levels of nestin) compared to static conditions. This phenomenon was attributed to the continuous nutrient, cytokine, and oxygen transport, as well as the removal of metabolic wastes provided by the presence of slow interstitial flow. Additionally, this system was utilized to investigate the toxic effects of amyloid-β, which is thought to be the main cause of Alzheimer’s disease. Under flow conditions, the results revealed a decreased viability of neurospheroids, as well as increased neural destruction and synaptic dysfunction compared to static conditions. Conclusively, this system had significant potential as an *in vitro* brain model since it offered an environment that was similar to that found *in vivo* (Figure 8J,K).

The neuronal response to perfusion and shear stress through microfluidic systems is summarized in Table 3.

### 6.3. Microfluidic Cell Co-Culture Platforms

Majumdar et al. [108] designed a PDMS microfluidic cell co-culture platform that allowed individual manipulation of various cell types with the placement of a microfabricated valve that served as a reversible barrier between the chambers. As a result, healthy co-cultures of hippocampal neurons and glia were maintained for several weeks under optimal conditions. In particular, co-culture with glia provided nutrient media for maintaining healthy neural cultures, eliminating the need to supply neurons with pre-conditioned glia media, thereby enhancing the transfection efficiency of neurons in the platform (Figure 9A–C). Similarly, Shi et al. [109] fabricated two PDMS microfluidic cell culture systems, a vertically-layered set-up, and a four-chamber set-up for studying communication between neurons and glia in close proximity. The chambers were separated by pressure-enabled valve barriers that allowed them to control communication between the two cell types. In this study, the number and stability of synaptic contacts, as well as the secreted levels of soluble factors, were increased in the co-culture system, thus confirming the importance of communication between neurons and glia for the development of stable synapses in microfluidic platforms (Figure 9D,E). Robertson et al. [110] developed an *in vitro* system to examine the synaptic interaction between two interconnected populations of mixed primary hippocampal co-cultures by integrating microfluidics with calcium imaging techniques. Moreover, a computational model was verified to characterize the fluidic characteristics of the system and improve the experimental protocols (prevent substance cross-contamination between co-cultures). The results revealed that neurons and glia, in each of the separated chambers, grew within the microchannels where they physically interacted and formed synapses. In addition to this, the function of the neuron-glia synapse was confirmed by calcium imaging (Figure 9F,G). Yang et al. [111] fabricated a microfluidic array platform to modulate hNSC differentiation in a 3D ECM microenvironment using recapitulation of paracrine action of genetically engineered human mesenchymal stem cells (hMSCs). hMSCs were genetically engineered to increase the expression of glial cell-derived neurotrophic factor (GDNF) utilizing cationic polymer nanoparticles. A simulation study produced by mathematical modeling of accumulated GDNF secreted from engineered hMSCs confirmed that *in vivo*-like signaling of secreted factors could be created in a 3D ECM hydrogel in the microfluidic system. Specifically, in the central channels of the microfluidic system, which were filled with a 3D ECM hydrogel, hNSCs were cultivated, and GDNF-overexpressing hMSCs (GDNF-hMSCs) were cultured in the channels on each side of the central channel. Reduced differentiation of hNSCs into glial cells and increased differentiation of hNSCs into neuronal cells, including dopaminergic neurons, were observed in the co-culture of hNSCs with GDNF-hMSCs in the 3D microfluidic system. Moreover, neuronal cells demonstrated functional neuron-like electrophysiological characteristics. Finally, an animal model of hypoxic-ischemic brain injury was used to confirm the improved paracrine capacity of GDNF-hMSCs (Figure 9H,I). Park et al. [112] fabricated a circular microfluidic co-culture platform where embryonic CNS neurons and postnatal oligodendrocytes (OLs) were co-cultured in two separated compartments which were connected by arrays of axon-guiding microchannels. These microfluidic channels allowed physical isolation for cell bodies, but not for axons, and maintained fluidic isolation. The embryonic CNS neurons and postnatal OL progenitors were co-cultured in the platform for up to four weeks to study the axon-glia interaction and myelination. The circular design demonstrated excellent cell loading characteristics where a significant number of cells were positioned near the axon-guiding microchannels. Moreover, it showed enhanced axonal growth characterized by the significantly increased axon coverage ratio in the axon-glia compartment. The co-culture capability of the platform was confirmed by successfully co-culturing OL progenitors with axons in the axon/glia compartment resulting in the maturation of OLs (Figure 9J,K). In a related study, Park et al. [113] developed a multi-compartment microfluidic co-culture platform for studying axon-glia interaction at a higher throughput. The platform allowed for conducting parallel localized biomolecule and drug treatments while carrying out various co-culture conditions in a single device. Using this platform, they were able to simultaneously study the axon-glia communication, the development and differentiation of oligodendrocytes, as well as the axonal-specific response to different stimuli. The results revealed that mature oligodendrocytes were needed in order to obtain a robust myelin sheath instead of oligodendrocyte progenitor cells. Additionally, it was shown that astrocytes stimulated the development of oligodendrocytes and were detrimental when added to an already existing axonal layer (Figure 9L,M). Finally, Ristola et al. [114] developed a novel PDMS-based microfluidic cell culture device, with open compartments, for neuron-oligodendrocyte *in vitro* myelin studies. It was shown that the primary rat DRG neurons were successfully co-cultured with the oligodendrocytes in the device, which could also be used for time-lapse imaging. The results also demonstrated successful interactions and contacts between neurites and oligodendrocytes, as well as the deposition of myelin segments in an aligned distribution in the device (Figure 9N,O).

Table 4 summarizes the co-culture studies of neural cells using microfluidic systems.

## 7. Application of Microfluidic Systems in Combination with Topography for Neural Studies

It has been well reported, as previously described in Section 3, that surface topography significantly affects the adhesion, orientation, proliferation, and differentiation of cells. Therefore, many studies have focused on the controlled modification of materials’ surfaces, with the ultimate aim of guiding neuronal outgrowth, which is considered crucial for the development of functional neuronal interfaces [3,4,5,6,30,33,37,40,41,43,115,116,117,118,119,120,121,122,123,124,125,126,127,128,129,130,131,132,133,134,135,136,137,138,139,140,141,142,143,144,145]. Nevertheless, the combined effect of microfluidic flow and topography on neuronal outgrowth has been rarely reported.

Specifically, Hesari et al. [146] fabricated a hybrid microfluidic system composed of a PDMS microchip and poly(lactic-co-glycolic acid) (PLGA) nanofiber-based substrate for differentiation of human induced pluripotent stem cells (hiPSCs) into neurons. The results revealed an increase in β-tubulin III (a specific neuronal marker) gene expression and a decrease in glial fibrillary acidic protein (GFAP) (a classic astrocyte marker) gene expression. Thus, this hybrid microfluidic system could be optimum for neuronal differentiation. Afterward, for further *in vivo* evaluation of this system, the cell-loaded scaffold was implanted in a spinal cord model of rats. During the 28 days of study, animals who received this implant displayed improved functional outcomes. However, the difference with the control group was not statistically significant (Figure 10A–D).

Kim et al. [1] developed a fluid flow system to study the effect of mechanical stimulation on PC12 cells cultured in microfiber-based substrates. They observed that the shear stress affected the length and orientation of neurons along the microfibers (Figure 10E–H). Furthermore, Jeon et al. [147] investigated the combined effects of topography and flow-induced shear stress on the neuronal differentiation of hMSCs. They applied different shear stresses in a PDMS substrate with micrometric grooves. The results demonstrated that shear stresses affected the expression of synaptophysin, β-tubulin III, and microtubule-associated protein 2 (MAP2), as well as the intracellular calcium concentration. The alignment was also confirmed. In addition to this, an increased neurite length was noticed on the seventh day. However, a significant decrease in neurite length was observed on the tenth day (Figure 10I–L). It should be noted that in both studies [1,147], the flow was not continuous but was rather applied for only a few hours per culture day.

Babaliari et al. [148] have studied the combined effect of shear stress and topography on Schwann (SW10) cells’ behavior under dynamic culture conditions attained via continuous flow. For this purpose, a precise flow-controlled microfluidic system with specific custom-designed chambers incorporating laser-microstructured polyethylene terephthalate (PET) substrates comprising microgrooves [36] was developed. The microgrooves were positioned either parallel or perpendicular to the direction of the flow and the response of SW10 cells was evaluated in terms of growth, orientation, and elongation. The cell culture results were also combined with computational flow simulation studies employed to accurately calculate the shear stress values. It was revealed that, depending on the relation of the direction of the flow with respect to the topographical features, wall shear stress gradients were acting in a synergistic or antagonistic manner to topography in promoting guided morphologic cell response (Figure 10M–Q).

Table 5 summarizes the reported studies on the combined effect of microfluidic flow and topography on neuronal response.

## 8. Conclusions and Future Perspectives

Despite the fact that the PNS displays a higher rate of regeneration than that of the CNS through spontaneous regeneration after an injury, functional recovery is fairly misdirected and infrequent. Hence, the discovery of successful methods to guide neurite outgrowth, *in vitro*, is of high significance. Considering the limitations of the grafts that are currently available, a novel promising field of neural tissue engineering has emerged for the development of new nerve graft substitutes. The ultimate target of a tissue-engineered construct is to imitate as closely as possible the physiological environment and provide the required cues (e.g., topographic features of the ECM, shear stress, soluble factors) so that cells will respond within the artificial environment as they would *in vivo*. Various approaches have been developed to create structures on surfaces that aim to reconstruct the architecture of ECM *in vitro.* However, apart from topography, another crucial factor in neurogenesis that should not be neglected is the mechanical environment provided by the ECM (e.g., flow-induced shear stress). Microfluidic systems offer the possibility to recreate closely the cellular microenvironment through the various physicochemical and biomolecular stimuli that could provide. In this review, we present an overview of how topography and microfluidic flow affect neuronal behavior, either separately or in synergy, with the aid of innovative microfluidic systems. Moreover, we highlight the importance of using many different stimuli (e.g., topography, microfluidic flow, shear stress) to simulate the *in vivo* microenvironment more appropriately. Various microfluidic systems for neural studies are demonstrated focused on the gradient—chemical stimulation, perfusion—shear stress, and co-culture. In addition to this, microfluidic systems combined with topography for neural studies are presented. The studies reveal the necessity of developing *in vitro* biomimetic cell culture systems that enable the closer simulation of the *in vivo* microenvironment in order to promote neuronal outgrowth. However, microfluidic systems have not yet fully realized their potential in the field of neural regeneration. Indeed, although the effect of microfluidic flow on neuronal outgrowth has been studied thoroughly, the combined effect of microfluidic flow and topography is limited to a few studies. However, this combined effect can simulate better the actual complex environment *in vivo*, which includes cell–cell, cell–soluble factor, and cell–matrix interactions. In the future, we expect that more microfluidic systems combined with 3D culture substrates will be developed to better mimic the 3D environment *in vivo*. Using these systems, the guidance of neurite outgrowth, *in vitro*, will be achieved. This could be potentially useful in the field of neural tissue engineering with the development of autologous graft substitutes for nerve tissue regeneration.

## Figures and Tables

**Figure 1 bioengineering-10-00902-f001:**
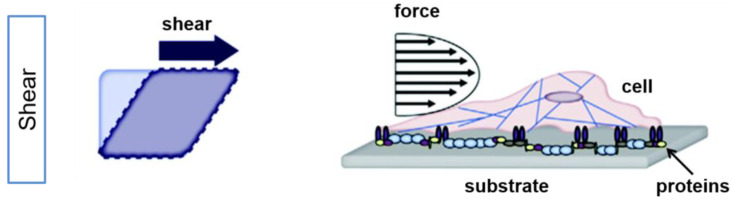
Externally applied force results in shear stress on cells. Simplified illustration of the effect of shear in planar culture is shown for an idealized square (**left**) and a cell (**right**). Shear stress induces an angle change between opposing sides of the cell (modified by [45]). Adapted with permission from [45].

**Figure 2 bioengineering-10-00902-f002:**
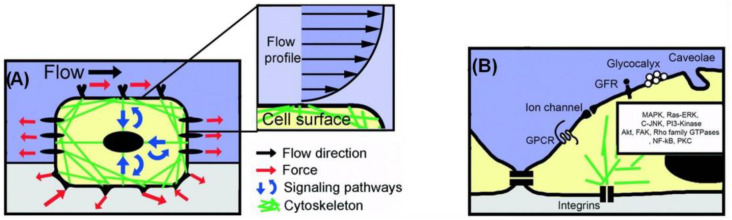
Cell mechanosensing model approach at the micro- and nano-scale. (**A**) A mechanosensing approach, where a cell internalizes both mechanical and biochemical signals. (**B**) Schematic illustration of selected membrane-bound nanoscale structures involved in intracellular mechanical signaling (modified by [47]). Adapted with permission from [47].

**Figure 3 bioengineering-10-00902-f003:**
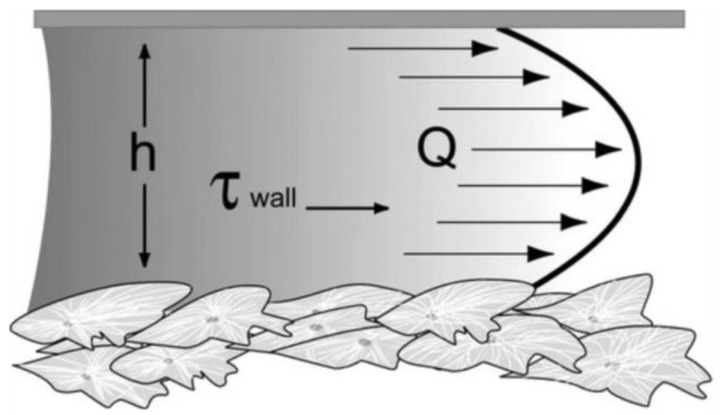
Fluid shear in a parallel-plate flow chamber. h refers to the separation (height) of the parallel plates, Q to the flow rate, and τ_wall_ to the wall shear stress [49]. Reprinted with permission from [49].

**Figure 4 bioengineering-10-00902-f004:**
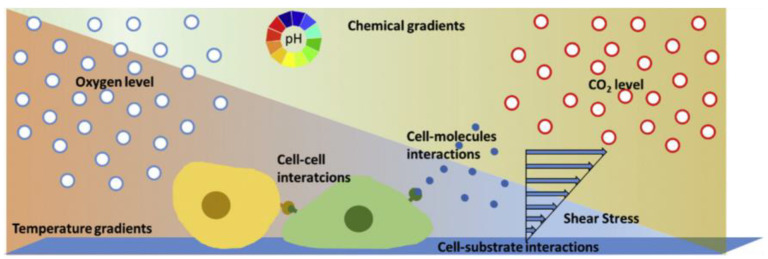
Schematic representation of the microenvironment of cells consisting of biochemical (cell interactions), physical (shear stress), and physicochemical (pH, carbon dioxide (CO_2_), temperature, oxygen (O_2_)) factors [15]. Reprinted with permission from [15].

**Figure 5 bioengineering-10-00902-f005:**
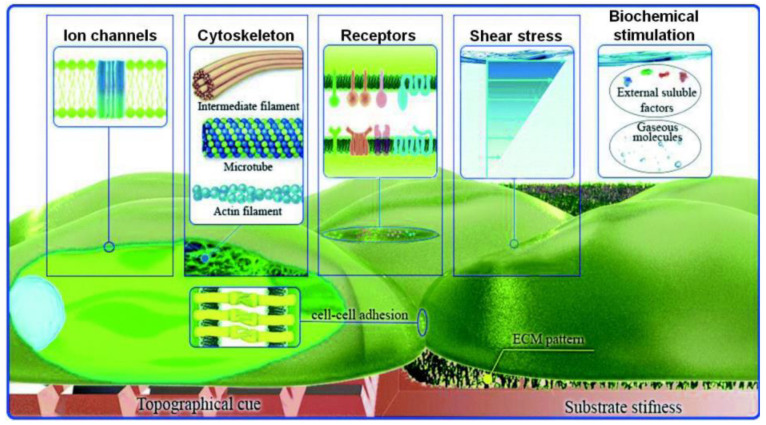
Schematic representation of the possible physicochemical and biomolecular stimuli, which could be provided by microfluidic flow (modified from [10]). Reprinted with permission from [10].

**Figure 6 bioengineering-10-00902-f006:**
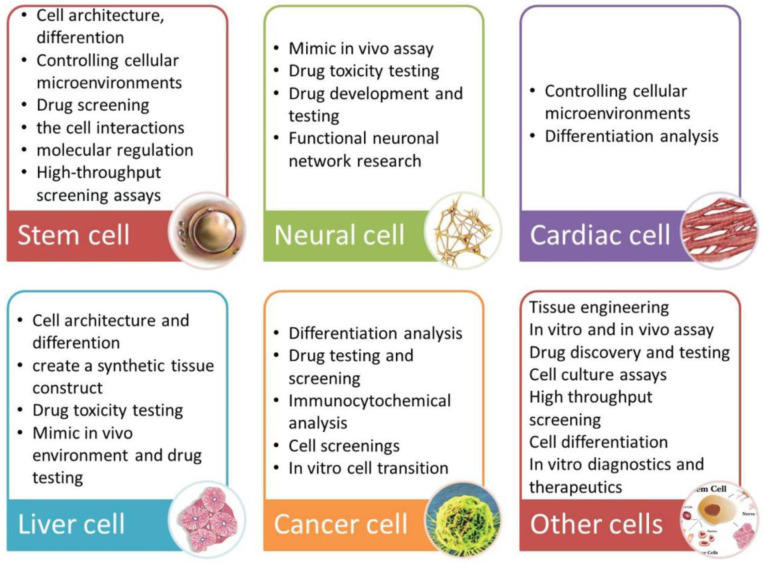
Summarized applications of cell culture under microfluidic flow according to cell type [95]. Reprinted with permission from [95].

**Figure 7 bioengineering-10-00902-f007:**
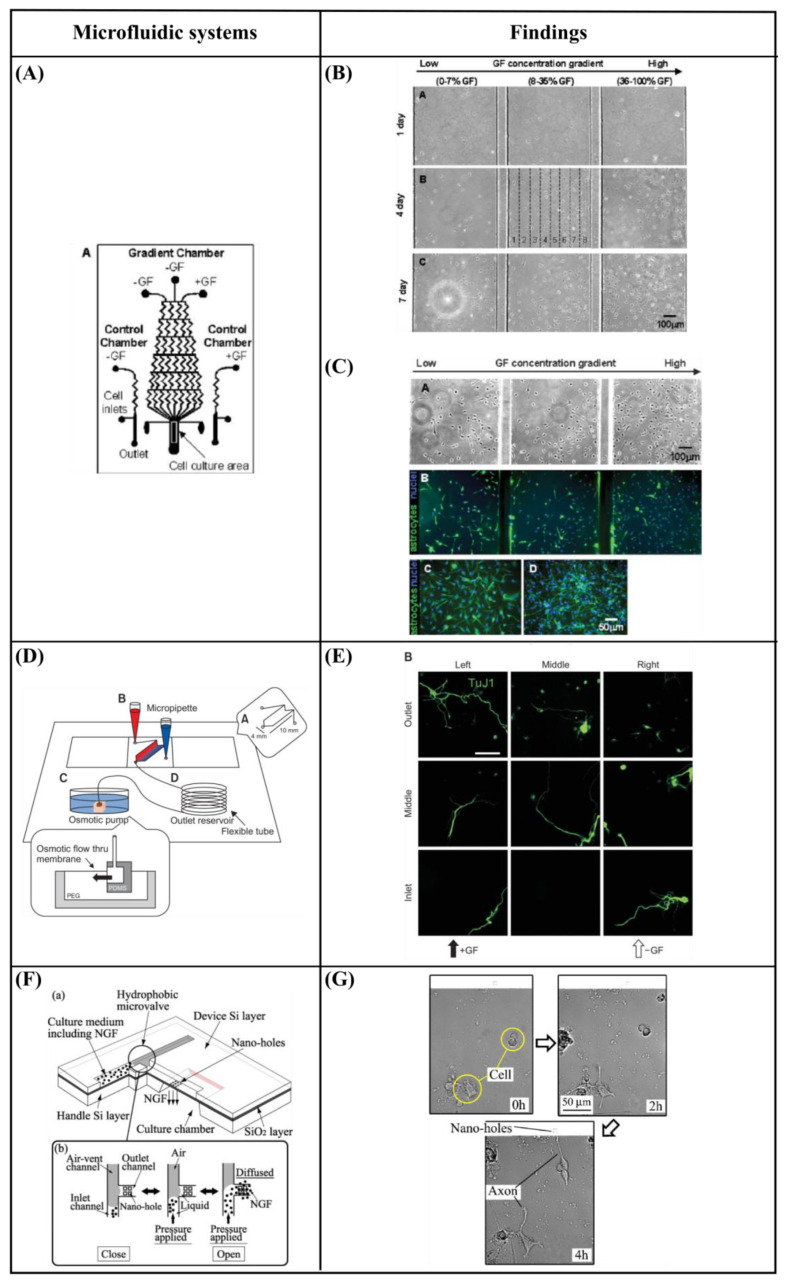
(**A**) Schematic design of the microfluidic device showing the gradient chamber and two control chambers [96]. (**B**) Proliferation of human neural stem cells (hNSCs) in the gradient chamber [96]. (**C**) Differentiation of hNSCs into astrocytes in the gradient chamber [96]. (**D**) Microfluidic chip device experimental device scheme [97]. (**E**) Immunocytochemistry data for neurons (TuJ1, green). 4′,6-Diamidino-2-Phenylindole (DAPI) was used to obtain the total number of cells. Scale bar = 50 μm [97]. (**F**) Schematic of a fabricated microfluidic device [98]. (**G**) Cells differentiation guidance on the fabricated microfluidic device [98]. (**H**) A microfluidic gradient generator: a polydimethylsiloxane (PDMS) slab with microchannel features and four holes bonded on a glass-bottom dish [99]. (**I**) Axon elongation of primary nerve (dorsal root ganglion (DRG)) cells from chick embryo in culture on the microfluidic chip [99]. (**J**) Experimental setup. The system consists of four different components [100]. (**K**) Quantification of neuronal cell body clusters and neurite bundles in the neuronal network generated with sonic hedgehog (Shh)-fibroblast growth factor 8 (FGF8) and Shh-bone morphogenetic protein 4 (BMP4) gradients. Rectangular mesh = 1300 × 670 μm [100]. (**L**) Microjet gradient array [101]. (**M**) Hippocampal neuron culture in gradient chambers [101]. (**N**) Schematic illustration of microfluidic chip fabrication process [102]. (**O**) Phase contrast microscope images of placenta-derived multipotent stem cells (PDMCs) under physical-chemical stimulation at various flow rates with/without 3-isobutyl-1-methylxanthine (IBMX) [102]. (**P**) Immunocytochemistry staining results after chemical and shear stress stimulation [102]. Reprinted with permission from [96,97,98,99,100,101,102].

**Figure 8 bioengineering-10-00902-f008:**
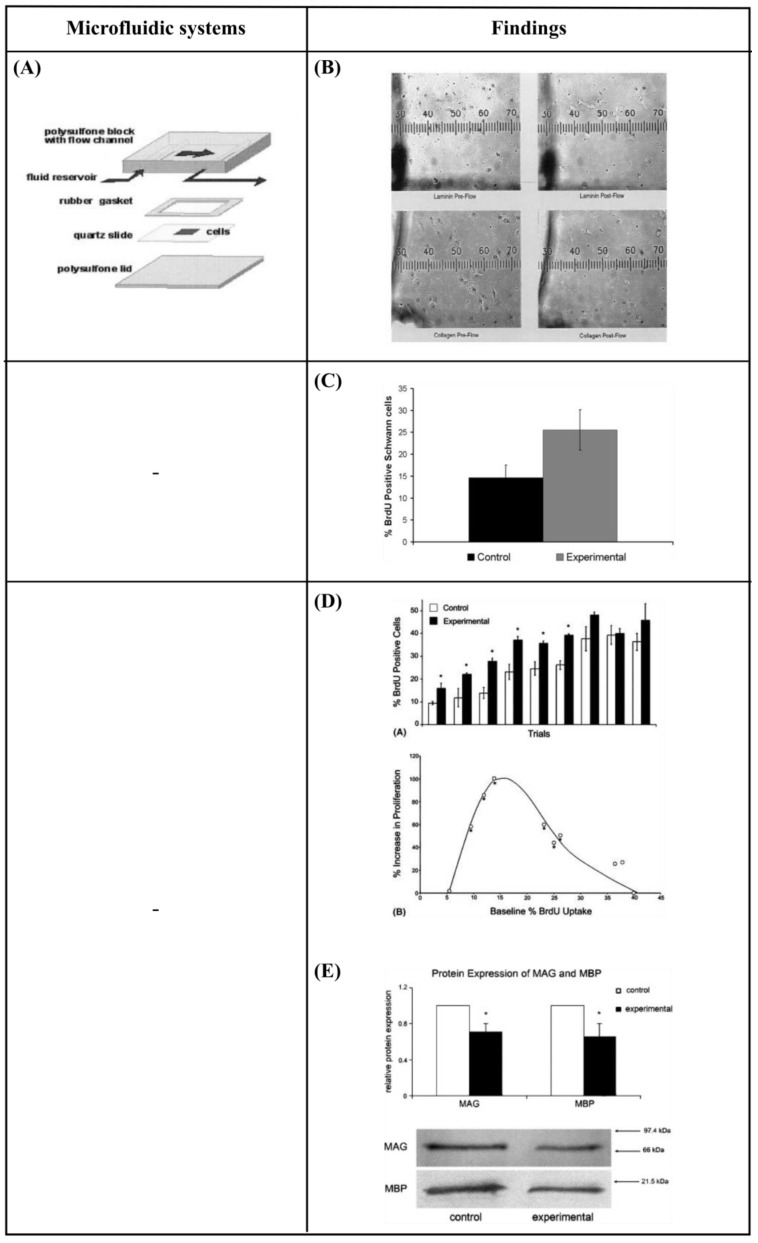
(**A**) Schematic diagram of a custom-designed flow chamber that was loaded with a substrate-coated and Schwann cell-seeded slide [9]. (**B**) Images of Schwann cells plated on slides coated with laminin and type IV collagen before and after the administration of laminar fluid flow [9]. (**C**) Data showing the percentage of *in vitro* Schwann cells that were *bromodeoxyuridine* (BrdU) positive after 2 h of mechanical stimulation with shear stress [103]. (**D**) Increase in proliferation of cultured Schwann cells in response to mechanical stress [104]. (**E**) Western blot results for myelin-associated glycoprotein (MAG) and myelin basic protein (MBP) are expressed relative to control protein levels with standard errors of the mean (SEM) [104]. (**F**) Schematic diagram of the fabrication of the microfluidic device [105]. (**G**) Neurons cultured in treated and untreated closed-channel polydimethylsiloxane (PDMS) microfluidic devices at 7 DIV with continuous, gravity-induced flow. Scale bar = 50 μm [105]. (**H**) Microfluidic chip design, fabrication, computational simulation, and operation [106]. (**I**) The effect of physiologically relevant shear stress on the proliferative potential of radial glial cells (RGCs) [106]. (**J**) Schematic diagrams of normal brain mimicking microfluidic chip (a) and Alzheimer’s disease brain mimicking microfluidic chip (b) [107]. (**K**) Neural network formation, average number of neurites extending from microwells, and average size of neurospheroids of group I and group II [107]. Reprinted with permission from [9,103,104,105,106,107].

**Figure 9 bioengineering-10-00902-f009:**
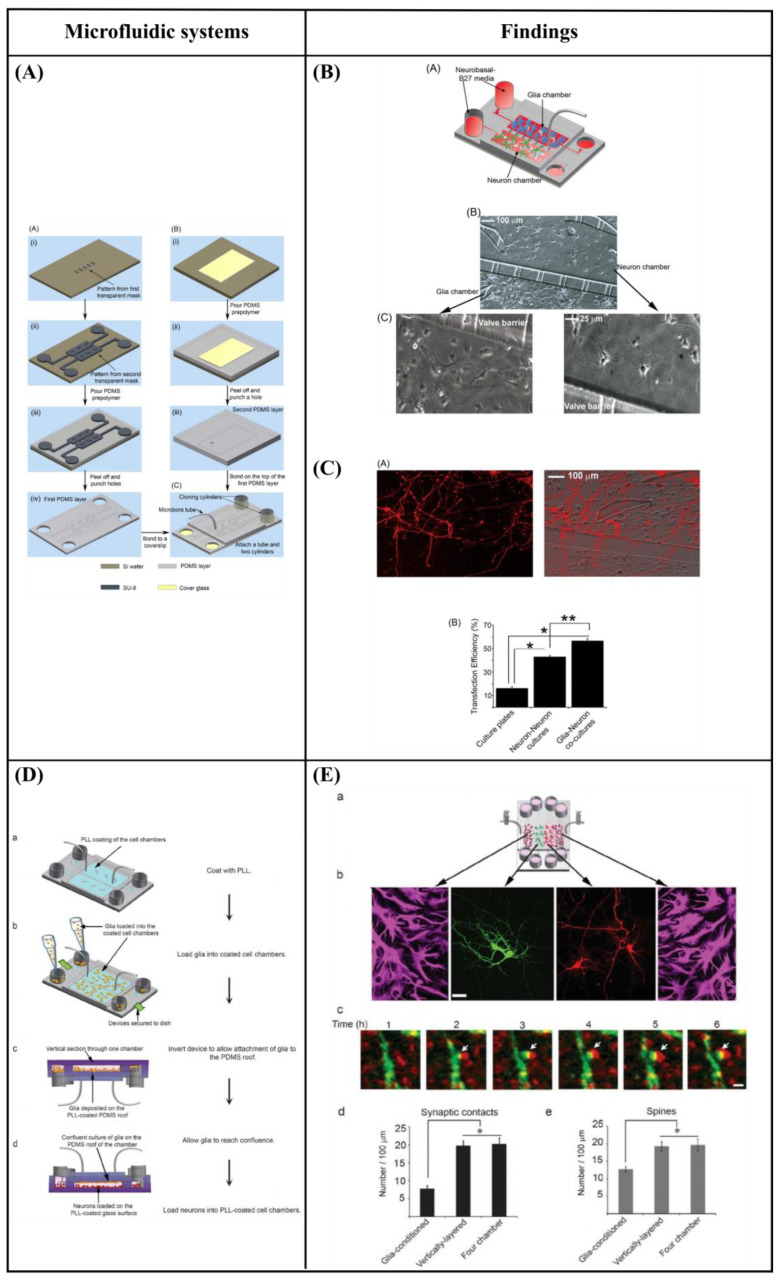
(**A**) Schematic of the microfluidic platform [108]. (**B**) Co-culture of neurons and glia in the microfluidic platform [108]. (**C**) Glia increase the transfection efficiency of neurons [108]. (**D**) Flow chart for cell loading in vertically-layered microfluidic platforms [109]. (**E**) Neuronal interaction within four chamber devices. Scale bars = 25 μm (b) and 2 μm (c) [109]. (**F**) Device structure and fluorescent microscopy setup [110]. (**G**) Neuronal and astrocytic processes project within the microchannels [110]. (**H**) Microfluidic platform for mimicking paracrine signaling of mesenchymal stem cells (MSCs) that controls neural stem cell (NSC) differentiation *in vivo* [111]. (**I**) Enhanced dopaminergic neuronal differentiation of human neural stem cells (hNSCs) co-cultured with glial cell-derived neurotrophic factor (GDNF)-overexpressing human mesenchymal stem cells (hMSCs) for 7 days in the three-dimensional (3D) microfluidic device. Scale bar = 50 μm [111]. (**J**) Schematic illustration of the microfluidic compartmentalized central nervous system (CNS) neuron co-culture platform [112]. (**K**) A phase contrast image and immunocytochemistry images of axons and oligodendrocytes (OLs) co-cultured inside the axon/glia compartment for two weeks. Scale bars = 20 μm [112]. (**L**) 3D illustration of the multi-compartment neuron-glia co-culture microsystem capable of carrying out multiple localized axon treatments in parallel [113]. (**M**) Images showing co-cultured axons and glial cells at DIV 27. Scale bars = 100 μm [113]. (**N**) Schematic illustration of the compartmentalized cell culture device for neuron-oligodendrocyte co-culturing [114]. (**O**) Myelination in the microfluidics device. Scale bars = 20 μm [114]. Reprinted with permission from [108,109,110,111,112,113,114].

**Figure 10 bioengineering-10-00902-f010:**
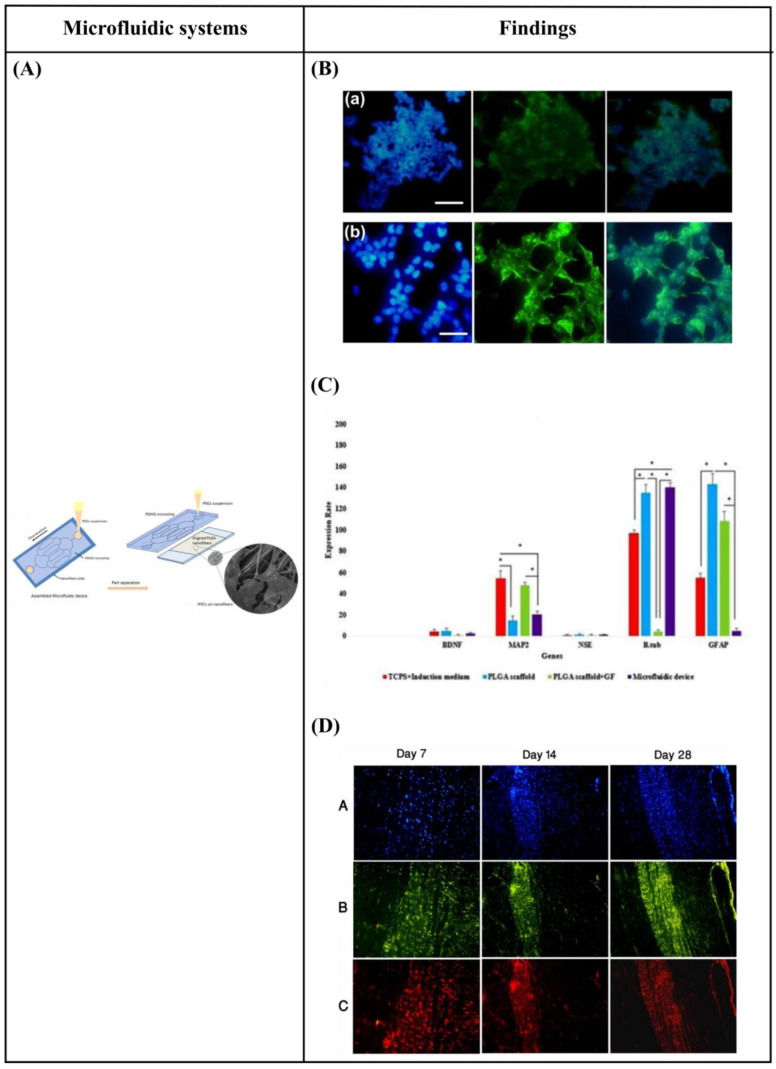
(**A**) Schematic representation of assembled and separated parts of hybrid device for stem cell loading and differentiation [146]. (**B**) The cells were subjected to immunocytochemistry analysis for the expression of neural markers including β-tubulin III (green). Cells were co-stained with 4′,6-Diamidino-2-Phenylindole (DAPI) to visualize nuclei (blue) [146]. (**C**) Neural gene expressions in differentiated human induced pluripotent stem cells (hiPSCs) on different surfaces [146]. (**D**) Scaffold-loaded cells survival in hemisected rat spinal cord after surgery during 28 days of an animal study in a cell containing the scaffold group [146]. (**E**) The image above shows a schematic diagram of the fluid flow system. The images below show the schematics of the flow chamber used for applying shear stress [1]. (**F**) Changes in cell morphology depending on the magnitude of shear stress and surface morphologies at 4 d after seeding. Scale bars = 100 μm [1]. (**G**) Angular deviation of neurites on two types of substrate along with various shear stresses [1]. (**H**) Average outgrowth length of neurites on different substrates along with various shear stresses [1]. (**I**) Fluid flow system for shear stress [147]. (**J**) Immunofluorescence staining results of human mesenchymal stem cells (hMSCs) cultured on the 5 μm patterned substrate under shear stress at day 10 [147]. (**K**) Outgrowth length of neurites on the 5 μm patterned substrate exposed to different magnitudes of shear stress [147]. (**L**) Quantification of the percentage of cells at day 10 that responded to neuronal activators on the 5 μm patterned substrate treated with different shear stresses [147]. (**M**) (a) Schematic illustration of the custom-designed microfluidic system; (b) cross-section image of the chamber, containing the laser-microstructured substrates and cells, where the flow occurs [148]. (**N**) Confocal images of Schwann (SW10) cells cultured on the polyethylene terephthalate (PET)-Flat or inside the microgrooves (MG) of the PET-MG substrates, under static or dynamic conditions, applying 50 and 200 μL min^−1^, on the third day of culture [148]. (**O**) Cell length of SW10 cells on the PET-Flat and PET-MG substrates under static and dynamic conditions, applying 50 and 200 μL min^−1^, on the third day of culture [148]. (**P**) Contour plots on PET-Flat and PET-MG substrates and cross-section profiles (dotted line) of computed wall shear stress for 50 and 200 μL min^−1^ flow rates [148]. Reprinted with permission from [1,146,147,148].

**Table 1 bioengineering-10-00902-t001:** Differences between conventional static cultures and cell cultures under microfluidic flow.

Conventional Static Cultures	Cell Cultures under Microfluidic Flow
Non-confined systems	Confined systems
No fluid flow	Fluid flow (e.g., laminar, turbulent)
Static nutrient and waste products	Consistent nutrient delivery and effective waste removal
Large volume of nutrient/reagents	Small volume of nutrient/reagents
Limited possibilities for creating mechanical stimulation	Mechanical stimulation possible (e.g., shear)
Limited level of spatial control	High level of spatial control
Low spatial and temporal control of chemical stimuli	High spatial and temporal control of chemical stimuli
Limited possibilities for integration and parallelization	High integration and parallelization
Low possibility for in situ read-out of biological processes	Possibility to integrate assays and sensors for in situ read-out of biological processes
Simple	Complex experimental set-up
Easy to handle	Complex operational control
Established culture protocols	Non-standard culture protocols
Compatibility with conventional biological assays	Compatibility issues with conventional biological assays
Compatibility with established read-out equipment	Compatibility issues with established read-out equipment

**Table 2 bioengineering-10-00902-t002:** Reported studies on the gradient and chemical stimulation through microfluidic systems on nerve cell morphology/response.

Gradient Generation—Chemical Stimulation				
Cell Type	Circulating Flow?	Shear Stress	Findings	Ref.
Human neural stem cells (hNSCs)	Yes	5 × 10^−5^ Pa for 4 and 7 days	▪ Minimization of autocrine and paracrine signals ▪ Differentiation of hNSCs into astrocytes was inversely proportional to growth factors (GF) concentration ▪ Proliferation was directly proportional to GF concentration ▪ Simultaneous application of multiple gradients in a single experiment ▪ Low media requirements ▪ Low cell requirements	[96]
Neural stem cells (NSCs)	Yes	-	▪ Proliferation and differentiation of NSCs were directly dependent on GF concentration	[97]
Adrenal pheochromocytoma (PC12) cells	No	-	▪ Cell growth, differentiation and axon elongation were dependent on micro-valve switching and release gradient of nerve growth factor (NGF)	[98]
Primary dorsal root ganglion (DRG) neuronal cells	No	-	▪ Directional elongation of axons following the NGF concentration gradient	[99]
Human embryonic stem cells (ESCs)-derived neural progenitor cells	Yes	-	▪ Proliferation and differentiation of neural progenitor cells in a controlled manner ▪ Cell properties reflected the different concentrations of extracellular signaling molecules	[100]
Primary hippocampal neurons	Yes	Lower than 1.2 × 10^−3^ Pa	▪ Hippocampal axon guidance was dependent on the concentration and incidence angle of netrin-1 gradient	[101]
Placenta-derived multipotent stem cells (PDMCs)	Yes	▪ 1.4 × 10^−4^ Pa ▪ 3.31 × 10^−3^ Pa ▪ 4.97 × 10^−3^ Pa for 1, 2, 24, 48, 72 h	▪ No differentiation of PDMCs into other cell types with shear stress ▪ Shear stress combined with 3-isobutyl-1-methylxanthine (IBMX) enhanced the PDMCs for earlier neuronal differentiation ▪ Highest cell differentiation in the maximum flow rate and IBMX in 48 h condition	[102]

**Table 3 bioengineering-10-00902-t003:** Reported studies on the perfusion and shear stress through microfluidic systems on nerve cell morphology/response.

Perfusion—Shear stress				
Cell Type	Circulating Flow?	Shear Stress	Findings	Ref.
Schwann cells	Yes	1.33 Pa for 2 h	▪ Increased proliferation ▪ Slight cells’ movement from their original positions	[9]
Schwann cells	Yes	3.1 Pa for 2 h	▪ Increased proliferation	[103]
Schwann cells	Yes	3.1 Pa for 2 h	▪ Increased proliferation ▪ Down-regulation of myelin-associated glycoprotein (MAG) and myelin basic protein (MBP)	[104]
Primary hippocampal neurons	No	-	▪ Increased viability and channel-length capacity	[105]
Radial glial cells (RGCs)	Yes	▪ 10^−4^ Pa ▪ 10^−3^ Pa for 5 days	▪ Increased proliferation of RGCs in response to increased shear stress	[106]
Neural progenitor cells	Yes	-	▪ Larger size of neurospheroids ▪ Formation of more complex and robust neural networks ▪ Increased levels of synapsin IIa and β-ΙΙΙ tubulin ▪ Decreased levels of nestin ▪ Investigation of the toxic effects of amyloid-β: decreased viability of neurospheroids; increased neural destruction and synaptic dysfunction	[107]

**Table 4 bioengineering-10-00902-t004:** Reported studies on the co-culture of neural cells using microfluidic systems on nerve cell morphology/response.

Co-Culture				
Cell Type	Circulating Flow?	Shear Stress	Findings	Ref.
Co-culture of hippocampal neurons and glia	No	-	▪ Maintenance of healthy co-cultures for several weeks under optimal conditions ▪ Elimination of the need to supply neurons with pre-conditioned glia media ▪ Enhancement of the transfection efficiency of neurons	[108]
Co-culture of hippocampal neurons and glia	No	-	▪ Increased number of synaptic contacts ▪ Increased stability of synaptic contacts ▪ Increased secreted levels of soluble factors	[109]
Primary hippocampal co-culture of neurons and glia	No	-	▪ Growth of neurons and glia within the microchannels ▪ Formation of synapses ▪ Confirmation of the function of neuron-glia synapse via calcium imaging	[110]
Co-culture of human neural stem cells (hNSCs) with glial cell-derived neurotrophic factor (GDNF)-overexpressing human mesenchymal stem cells (hMSCs)	No	-	▪ Reduced glial differentiation of hNSCs ▪ Enhanced differentiation of hNSCs into neuronal cells including dopaminergic neurons ▪ Functional neuron-like electrophysiological features of neuronal cells ▪ Confirmation of the enhanced paracrine ability of GDNF-hMSCs via an animal model of hypoxic-ischemic brain injury	[111]
Co-culture of embryonic central nervous system (CNS) neurons and postnatal oligodendrocytes (OLs)	No	-	▪ Enhanced axonal growth; increased axon coverage ratio in the axon-glia compartment ▪ Maturation of OLs	[112]
Co-culture of CNS neuron and glia	No	-	▪ To obtain a robust myelin sheath mature oligodendrocytes were needed instead of oligodendrocyte progenitor cells ▪ Stimulation of the development of oligodendrocytes by astrocytes ▪ Detrimental effect of astrocytes when added to a pre-existing axonal layer	[113]
Co-culture of rat dorsal root ganglion (DRG) neurons and oligodendrocytes	No	-	▪ Successful interactions and contacts between neurites and oligodendrocytes ▪ Deposition of myelin segments in an aligned distribution	[114]

**Table 5 bioengineering-10-00902-t005:** Reported studies on the combined effect of microfluidic flow and topography on nerve cell morphology/response.

Substrate	Fabrication Technique	Cell Type	Circulating Flow?	Shear Stress	Findings	Ref.
Poly(lactic-co-glycolic acid) (PLGA) nanofiber-based substrate	Electrospinning	Human induced pluripotent stem cells (hiPSCs)	No	-	▪ Increase in β-tubulin III gene expression ▪ Decrease in glial fibrillary acidic protein (GFAP) gene expression ▪ Animals receiving this implant showed functional improvement but no significant difference with the control group	[146]
PLGA microfiber-based substrate	Electrospinning	Adrenal pheochromocytoma (PC12) cells	Yes	0.1–1.5 Pa for 2 h, 3 times per day for 2 days	▪ 0.25 Pa: increased neurite length ▪ 0.5 Pa: increased cellular alignment	[1]
Polydimethylsiloxane (PDMS) substrate with micrometric grooves	Photolithography	Human mesenchymal stem cells (hMSCs)	Yes	0.1 and 0.25 Pa for 3 h per day for 2 days	▪ Shear stresses affected the expression of synaptophysin, β-tubulin III, and microtubule-associated protein 2 (MAP2) ▪ Increased neurite length at day 7; decrease in neurite length at day 10 ▪ Higher calcium concentration under 0.1 Pa ▪ Shear stress of 0.1 Pa most effective	[147]
Polyethylene terephthalate (PET) microgrooved substrate	Ultrafast laser direct writing	Schwann (SW10) cells	Yes	0.04 and 0.15 Pa, continuous flow for 2 days	▪ Flow parallel to microgrooves’ length: enhancement of cells’ alignment; increased cell length by 11.7% (0.04 Pa) and 12.3% (0.15 Pa) compared to static culture conditions ▪ Flow perpendicular to microgrooves’ length: cells retained their orientation along the direction of microgrooves; decrease in cell length	[148]

## Data Availability

Not applicable.

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
