# Peer review of "Microfluidic Systems for Neural Cell Studies"

_bioengineering, 2023, doi:10.3390/bioengineering10080902_

Round 1
Reviewer 1 Report
The authors of the review aimed to investigate the effects of topography and microfluidic flow on neuronal behavior with the ultimate goal of providing evidence for the efficacy of microfluidic systems in promoting neuronal outgrowth, thereby addressing a crucial research gap in the field. While the topic of the rewiew is interesting, there are a few aspects to be improved before the paper can be accepted, as follows:
1. The paper lacks a comprehensive introduction that adequately introduces the topic at hand. It is essential for the authors to provide a clear and concise introduction to microfluidic systems and their relevance to neural cell studies. Additionally, an introductory section is necessary to provide an overview of the various topics discussed later in the paper, such as neuroregeneration, topography, shear stress, cell response under flow, and microfluidics design. The absence of such an introduction disrupts the flow of the paper and makes it difficult for readers to grasp the overall context.
2. Similarly, the background information provided in the paper is insufficient to provide readers unfamiliar with the topic with the necessary context. The authors should aim to clearly define the scope and focus of the review, providing a comprehensive background that sets the stage for the subsequent discussions.
3. It is customary for the introduction of a paper to present a summary of the previous work in the field and clearly state the objectives or research questions addressed by the review. However, this essential aspect is lacking in the present manuscript, and the authors should make an effort to rectify this issue.
4. The research questions or objectives of the review were not explicitly articulated, and it would greatly enhance the clarity of the paper if the authors clearly state their research questions or objectives, providing a guiding framework for the subsequent discussions and analysis.
5. It is essential to outline the methodology employed in approaching the identified research gap. In this case, the authors should describe the inclusion/exclusion criteria used for selecting the papers reviewed and provide a rationale for the exclusion of certain papers. A clear description of the methodology would enhance the transparency and rigor of the review.
6. There appears to be a lack of flow between the paragraphs at line 42, resulting in a disjointed narrative. The authors should ensure smooth transitions between paragraphs to maintain a coherent and cohesive structure.
7. There is a concern regarding the definition of shear force on lines 141-142, even if it is based on a reference [29]. The definition should incorporate both force and viscosity terms to accurately represent shear stress. It is recommended that the authors consult a more reliable and authoritative source, such as a book or a peer-reviewed journal article, to support their definition.
8. Figure 6 would benefit from improvements in its design, particularly the removal of the grey border line, which could be distracting or aesthetically inconsistent with the other figures.
9. Figures 7, 8, 9, and 10 pose challenges for readers due to the small size of the text within them. It would be more effective if the authors select a specific snippet from each image and describe it in the text, rather than presenting the entire image. This approach would enhance readability and comprehension.
10. It is advisable to reorganize sections 5 and 6 as a more centralized section called "Microfluidic System Design." This section should encompass discussions on the design principles, fabrication techniques, and materials used in microfluidic systems, while also highlighting the advantages and limitations associated with each type of microfluidic system.
11. It is recommended that the authors introduce a new section called "Analysis and Characterization," which would provide a detailed description of the techniques and tools used for analyzing and characterizing neural cells within microfluidic systems. This section should cover a range of methods, including microscopy, immunostaining, electrophysiology, and biochemical assays. Additionally, the advantages and challenges associated with performing analysis and characterization within microfluidic systems should be discussed.
12. Similarly, it is recommended that the authors include a new section titled "Control and Manipulation of Microenvironments." This section would delve into the methods and strategies employed for controlling and manipulating the microenvironment within microfluidic systems, encompassing topics such as flow control, temperature regulation, and chemical gradients. The impact of controlled microenvironments on neural cell behavior, differentiation, and interaction should also be thoroughly discussed.
13. The conclusion should explicitly state whether the objective of the review was achieved or not. As it stands, the conclusion mentions that using microfluidic systems will be advantageous for studying the mechanobiology of neural cells, a topic that was not adequately addressed earlier in the paper. To address this, the authors should either introduce the topic of mechanobiology of neural cells during the review or present a separate section discussing the mechanobiological aspects.
14. Given the significance of the topic discussed in this review, it would be highly beneficial for the authors to include a dedicated discussion section. This section could provide a deeper analysis and critical evaluation of the technologies presented, including their advantages, disadvantages, and overall progress in the field.
15. Similarly, considering the importance of the topic, it is recommended that the authors include a section on "Future Perspectives and Challenges." This section should discuss the current challenges and limitations within the field of microfluidic systems for neural cell studies, identify emerging trends, and propose potential future directions for research and development. Additionally, highlighting the key technical, biological, and translational challenges that need to be addressed to advance the field would significantly enhance the review's contribution.
It is sufficiently OK but flow can be improved.
Reviewer 2 Report
The Review by Babaliari et al. discusses the challenges associated with the regeneration of axons in the peripheral nervous system (PNS). Although axons in the PNS can regenerate spontaneously, the regeneration process is often unsuccessful due to incorrect guidance cues. Therefore, it is crucial to discover effective methods to promote neurogenesis in in vitro systems.
The use of microfluidic systems is proposed as a suitable approach because they mimic the in vivo environment of cells in tissues, including normal fluid flow, nutrient delivery, waste removal, and mechanical stimulation through fluid shear forces. Additionally, this review highlights the significant influence of topography on neuronal outgrowth, orientation, and differentiation. It demonstrates how the combination of topography and microfluidic flow can impact neuronal behavior and emphasizes the effectiveness of microfluidic systems in facilitating neuronal outgrowth.
I have however a few remarks:
- I think that short chapter on basics of microfluidics could be a good idea. The article assumes good level of understanding of microfluidics (chips, pumps, method of manufacturing). I think that introduction would be god for someone who is not very familiar with these techniques
- the quality of figure 4 should be increased
- Figure 7, Figure 8, Figure 9 and Figure 10. I understand the idea of these arrangements but I do think that they are very hard to read. The schematics, pictures and figures taken from the articles are very hard to read. It should be re-arranges, maybe some information can be moved to the ESI? Please make them easier to read and follow.
The english language is good, no major issues were found.
Round 2
Reviewer 1 Report
The manuscript has improve significantly. Figure 8 and 9 contains information that is not possible to interpret (very small letters).
Reviewer 2 Report
The manuscript can be accepted for publication in present form.